# Street images classification according to COVID-19 risk in Lima, Peru: a convolutional neural networks feasibility analysis

Rodrigo M Carrillo-Larco [1,2,3] Manuel Castillo-Cara [4]
Jose Francisco Hernández Santa Cruz[5]

[1]Department of Epidemiology and Biostatistics, School of Public Health, Imperial College London, London, UK
[2]CRONICAS Centre of Excellence in Chronic Diseases, Universidad Peruana Cayetano Heredia, Lima, Peru
[3]Universidad Continental, Lima, Peru
[4]Universidad Politécnica de Madrid, Madrid, Spain
[5]Independent Researcher, Edinburgh, UK

**Correspondence to**
Dr Rodrigo M Carrillo-Larco; rcarrill@ic.ac.uk

## ABSTRACT

**Objectives** During the COVID-19 pandemic, convolutional neural networks (CNNs) have been used in clinical medicine (eg, X-rays classification). Whether CNNs could inform the epidemiology of COVID-19 classifying street images according to COVID-19 risk is unknown, yet it could pinpoint high-risk places and relevant features of the built environment. In a feasibility study, we trained CNNs to classify the area surrounding bus stops (Lima, Peru) into moderate or extreme COVID-19 risk.

**Design** CNN analysis based on images from bus stops and the surrounding area. We used transfer learning and updated the output layer of five CNNs: NASNetLarge, InceptionResNetV2, Xception, ResNet152V2 and ResNet101V2. We chose the best performing CNN, which was further tuned. We used GradCam to understand the classification process.

**Setting** Bus stops from Lima, Peru. We used five images per bus stop.

**Primary and secondary outcome measures** Bus stop images were classified according to COVID-19 risk into two labels: moderate or extreme.

**Results** NASNetLarge outperformed the other CNNs except in the recall metric for the moderate label and in the precision metric for the extreme label; the ResNet152V2 performed better in these two metrics (85% vs 76% and 63% vs 60%, respectively). The NASNetLarge was further tuned. The best recall (75%) and F1 score (65%) for the extreme label were reached with data augmentation techniques. Areas close to buildings or with people were often classified as extreme risk.

**Conclusions** This feasibility study showed that CNNs have the potential to classify street images according to levels of COVID-19 risk. In addition to applications in clinical medicine, CNNs and street images could advance the epidemiology of COVID-19 at the population level.

## INTRODUCTION

In COVID-19 research, deep learning tools applied to image analysis (ie, computer vision) have informed the diagnosis and prognosis of patients through the classification of X-ray and computer tomography images of the chest.[1–3] These tools have helped practitioners treating COVID-19 patients.

---

**STRENGTHS AND LIMITATIONS OF THIS STUDY**

⇒ We used five images per bus stop and the outcome information was provided by an official government institution.
⇒ We leveraged on five well-known convolutional neural networks (transfer learning).
⇒ The analysis focused on street images from one city only.
⇒ Original data (street images) cannot be shared because of restricted access.

---

On the other hand, the application of computer vision to study the epidemiology of COVID-19 has been limited. One relevant example is the use of Google Street View images to extract features of the built environment and associate these with COVID-19 cases in the USA.[4] This work showed that unstructured and non-conventional data sources, such as street images, can deliver relevant information to characterise the epidemiology of COVID-19 at the population level.[4] In a similar vein, though not exclusively addressing COVID-19, other researchers have leveraged on street images to study health-related social inequalities,[5] air pollution,[6] walkability,[7] as well as the built environment and health outcomes.[8 9] These examples show the potential of computer vision for population health research, above and beyond its multiple applications in clinical medicine with diagnostic and prognostic models.

However, to the best of our knowledge, computer vision models to classify street images based on their COVID-19 risk do not exist. From a public health perspective, such models could be relevant to understand unique local features of the built environment related to high COVID-19 risk. In addition, these models could be applied to places where observed data are not available to

identify whether this place is at moderate or high risk of COVID-19 and inform potential interventions. This could be particularly helpful in low-income and middle-income countries where limited resources do not allow massive COVID-19 testing, leaving places with no observed information about the COVID-19 epidemiology, though the local epidemiology could be estimated based on available images or alternative sources.

In this pilot feasibility study, we aimed to ascertain whether a convolutional neural network (CNN) (deep learning) model could classify street images of bus stops according to their COVID-19 risk (binary outcome: moderate vs extreme risk) in Lima, Peru. We also aimed to understand what features of the images were most influential in the classification process.

## METHODS
### Study design
We used CNNs to study street images of bus stops and their surroundings in Lima, Peru. We implemented a classification model to classify the bus stops into two labels: moderate or extreme risk of COVID-19. We addressed a classification problem.

### Rationale
We used 5 images per bus stop covering 360° around the bus stop. Therefore, we targeted the bus stop and the surrounding area. We did not target the bust stop itself alone. The bus stop was the anchor for the outcome label (moderate or extreme risk of COVID-19) in the immediate surrounding area. It is unlikely that COVID-19 risk would be confined to the bus stop itself. Rather, the bus stop would be a proxy of the risk in the immediate nearby area.

We combined the five images before randomly splitting into the train, test and validation dataset. We used the function *train_test_split* which randomly splits the data with equal distribution of the target outcome. We did not condition the random split on the bus stops because we did not target the bus stop itself only. A random split would provide data to have different profiles of the built environment, green areas, bus stops and other street features relevant for the model to learn and classify according to COVID-19 risk.

We deemed this a pilot feasibility proof-of-concept study because we aimed to provide preliminary data on whether CNNs could classify street images according to COVID-19 risk. While there is evidence about CNNs being used for classification of X-rays and other clinical images for COVID-19 diagnosis,[1–3] there is less evidence on CNNs being used for population health and COVID-19. Future research could leverage on this idea with more images, classifying into multiple outcome labels and implementing more sophisticated networks.

Public health and epidemiological research usually rely on structured data sources such as health surveys and measurements from patients including samples such as blood. Unstructured data sources, such as images, are gaining attention in clinical medicine and have been used to develop diagnosis and prognostic models; however, the use of images, including street images, in public health and epidemiological research is limited. This work elaborates on this premise and on the current burden by COVID-19 and was conceived to study whether street images can ascertain the COVID-19 risk in the community. If successful, a deep learning model to classify street images according to COVID-19 risk could be used for disease surveillance, and to estimate the risk in places where observed data lack.

### Data sources
The labels (observed data) of the bus stops were downloaded from the website of the Authority for Urban Transport in Lima and Callao (*Autoridad de Transporte Urbano para Lima y Callao*, name in Spanish). This government office manages the public transportation service in Lima, and publishes a classification map in which all bus stops in Lima are set into four categories of COVID-19 risk: moderate<high<very high<extreme.[10] Although this is an official source of information from a government branch, details of how the bus stops were classified are not available; please, refer to the discussion section where we further elaborate on this caveat. In this pilot feasibility study, we only worked with the bus stops deemed as moderate (label 0) and extreme (label 1) risk of COVID-19. We used the classification profile released on 24 May 2021.[11] We conducted a pilot feasibility study considering two outcome labels only. This, because we aimed to ascertain whether our hypothesis was possible and lead to relevant results while studying, from a public health perspective, the most important outcomes signalling the extremes of the risk distribution. Developing a model to identify areas at moderate risk could signal places where restrictions can be relaxed or suspended. Similarly, developing a model to identify areas at extreme risk would signal places where restrictions should be kept or strengthened. Therefore, a model focusing on two labels only, where these labels represent the extremes of the risk distribution, would be relevant and provide actional evidence. Our study could demonstrate that CNNs could successfully classify street images according to COVID-19 risk, with not addition information such as number of cases or health determinants. This has not been studied before. Future work will leverage on this preliminary experience to develop a four-outcome model, using larger datasets and incorporating more sophisticated networks.

We used the location (longitude and latitude coordinates) of the bus stops to download their street images through the application programming interface (API) of Google Street View. That is, we downloaded all the images in one batch through the API, rather than each one at the time through the API or from the standard Google Street View website. For each bus stop (ie, from each coordinate), we downloaded five images: when the camera was facing at 0°, at 90°, at 180° and at 270°; in addition, we

also downloaded one image in which the direction of the camera was not specified (ie, the heading parameter in the API request was set at default). In other words, for each bus stop we had five images. We did this to maximise the available data and to cover the surrounding area of the bus stop.[12] Our rationale was that the bus stop itself would not be responsible for the classification (moderate or extreme risk), but the whole nearby environment. Consequently, if the bus stop was labelled as moderate or extreme risk, the same label applied to the images of the surrounding area. For example, if bust stop X was labelled as moderate risk, all five images for such bus stop were labelled as moderate risk (ie, image of the bus top itself plus the four images of the surrounding area).

### Original dataset

Overall, after downloading both the labels and the images, there were 1788 bus stops with their corresponding label: 1173 in the moderate category and 615 in the extreme category (1173+615 = 1788). Because we used 5 images per bus stop, the analysis included 8940 (1788×5) images and their corresponding label. The training dataset included a random sample of 60% (5364) of the original dataset. As further explained in the next section (data preparation and class imbalance), after correcting for class imbalance by introducing duplicates of the class with fewer observations, the training data included 7024 observations (3519 for moderate and 3505 for extreme labels). The validation and test datasets included a random sample of 20% of the original dataset each (0.20×8940=1788); the validation and test datasets were not corrected for class imbalance.

### Data preparation and class imbalance

We combined the images and the labels in one dataset, which was further divided into three datasets: the training dataset including 60% of the data, the validation dataset including 20% and the test dataset including the remaining 20%. Data allocation to each of these three datasets was at random. After splitting the data, we corrected for class imbalance in the train dataset only. We randomly multiplied the number of images in the imbalanced outcome by 0.9. This led to virtually the same number of images for the moderate and extreme risks labels.

There were two outcomes of interest: moderate and extreme risk. However, there were more observations in the moderate category than in the extreme category. That is, there was class imbalance. After splitting the data into the training, test and validation sets, we corrected for class imbalance in the training dataset only. We randomly increased the number of observations in the extreme category by 90% in the training dataset (not in validation and test datasets). The original (before correction for class imbalance) training set had 3519 observations in the moderate category and 1845 in the extreme category (3519+1845=5364). After correcting for class imbalance as described before, the training dataset had 3519 observations in the moderate category (this number did not

change) and 3505 (1.9×1845) observations in the extreme category. Therefore, there were 3519 (moderate)+3505 (extreme after class imbalance correction)=7024 images and labels in total in the training dataset.

### Analysis

In-depth details about the analysis are available in online supplemental materials pp. 03–06. The analysis code (Python Jupyter notebooks) is also available in online supplemental materials.

In brief, in a prespecified protocol we decided to elaborate on five deep CNNs pretrained with ImageNet (ie, transfer learning). We chose these five networks because they have the best top five accuracy of all models available in the Keras library[12]: NASNetLarge, InceptionResNetV2, Xception, ResNet152V2 and ResNet101V2. We implemented these five models with the same hyperparameters, and then we selected the one with the best performance which was further tuned and tested. The image classification model was based on the latter model only (ie, the one with the best performance out of the five candidate models). We reported the loss and accuracy in the validation and test datasets; we also used the test dataset to report the accuracy, recall and F1 score for each of the two possible outcomes (moderate or extreme risk). Finally, we used GradCam (class activation maps) to identify which areas of the input image were more relevant to inform the classification process[13]; for this, we randomly selected four images per outcome (ie, four images from the moderate label and four images from the extreme label). Areas most activated as shown by brighter colours, would be decisively in the classification process.

### Patient and public involvement

Human subjects did not participate nor were involved in this study.

## RESULTS

### Selection of the pretrained model out of five candidate models

We used transfer learning and updated the output layer of five CNNs to predict our two classes of interest. The NASNetLarge architecture and weights outperformed the other candidate CNNs, except in the recall metric for the moderate label: 76% vs 85% in NASNetLarge and ResNet152V2, respectively, (table 1). The ResNet152V2 also performed better than the NASNetLarge in the precision metric for the extreme label (60% vs 63%). Further experiments were only conducted with NASNetLarge because, overall, it performed better than the other pretrained networks.

### Model performance

We further tuned NASNetLarge with different hyperparameters aiming to improve the accuracy (table 2).

First, building on the initial hyperparameters, we implemented two data augmentation options: horizontal flip and zoom range. We chose these two data augmentation

**Table 1** Performance of the five candidate convolutional neural networks

| | NASNetLarge | InceptionResNetV2 | Xception | ResNet152V2 | ResNet101V2 |
|---|---|---|---|---|---|
| Loss, validation | 0.526799 | 0.554040 | 0.533278 | 0.793147 | 0.744385 |
| Accuracy, validation | 0.742046 | 0.713636 | 0.730682 | 0.721023 | 0.723295 |
| Loss, test | 0.539906 | 0.557637 | 0.555917 | 0.800661 | 0.726274 |
| Accuracy, test | 0.731818 | 0.721591 | 0.706818 | 0.722727 | 0.718750 |
| Precision, label 0 (moderate) | 0.82 | 0.78 | 0.82 | 0.76 | 0.80 |
| Recall, label 0 (moderate) | 0.76 | 0.81 | 0.71 | 0.85 | 0.76 |
| F1 score, label 0 (moderate) | 0.79 | 0.79 | 0.76 | 0.80 | 0.78 |
| Precision, label 1 (extreme) | 0.60 | 0.61 | 0.56 | 0.63 | 0.58 |
| Recall, label 1 (extreme) | 0.68 | 0.56 | 0.70 | 0.48 | 0.64 |
| F1 score, label 1 (extreme) | 0.64 | 0.58 | 0.62 | 0.54 | 0.61 |

Green colour highlights the best metric, yellow colour highlights the second best metric and red colour highlights the third best metric row-wise. The precision, recall and F1 score are presented as proportions (multiply by 100 to have percentages). The precision, recall and F1 score were computed with the test dataset. Receiver operating characteristic curves for each model are available in online supplemental materials.

methods because they appropriately fit the images under analysis; for example, because we were working with street images, a vertical flip would not seem appropriate. The new model with horizontal flip improved the recall and F1 score for the extreme label; from 68% with the original NASNetLarge to 75%, and from 64% to 65% (figure 1). The new model with horizontal flip and zoom range at 30% had better performance than the original NASNetLarge model in 6 out of 10 parameters, including precision for the extreme label.

Second, also building on the initial hyperparameters (ie, without data augmentation), the decay in the stochastic gradient descendent optimiser was changed from 1/25 (25 was the number of epochs) to 1/10 (the number of epochs was not changed). This model did not substantially improve the performance of the model.

**Table 2** Further tuning of the selected model (NASNetLarge) and the performance metrics

| | Original model (as in table 1) | New model specifications | | | |
|---|---|---|---|---|---|
| | | horizontal_flip=True// epochs=25 (stopped at 12 epochs) | horizontal_flip=True// zoom_range=0.30// epochs=25 (stopped at 15 epochs) | decay=0.1/10// epochs=25 (stopped at 12 epochs) | decay=0.1/10// factor=0.3// epochs=25 (stopped at 12 epochs) |
| Loss, validation | 0.526799 | 0.534797 | 0.537553 | 0.532246 | 0.532246 |
| Accuracy, validation | 0.742046 | 0.737500 | 0.739773 | 0.732386 | 0.732386 |
| Loss, test | 0.539906 | 0.550286 | 0.528204 | 0.538252 | 0.538252 |
| Accuracy, test | 0.731818 | 0.719318 | 0.735795 | 0.725568 | 0.725568 |
| Precision, label 0 (moderate) | 0.82 | 0.85 | 0.76 | 0.83 | 0.83 |
| Recall, label 0 (moderate) | 0.76 | 0.71 | 0.87 | 0.74 | 0.74 |
| F1 score, label 0 (moderate) | 0.79 | 0.77 | 0.81 | 0.78 | 0.78 |
| Precision, label 1 (extreme) | 0.60 | 0.57 | 0.66 | 0.59 | 0.59 |
| Recall, label 1 (extreme) | 0.68 | 0.75 | 0.47 | 0.71 | 0.71 |
| F1 score, label 1 (extreme) | 0.64 | 0.65 | 0.55 | 0.64 | 0.64 |

Green colour highlights the best metric, yellow colour highlights the second best metric and red colour highlights the third best metric row-wise considering only the new model specifications. The precision, recall and F1 score are presented as proportions (multiply by 100 to have percentages). receiver operating characteristic curves for each model are available in online supplemental materials.

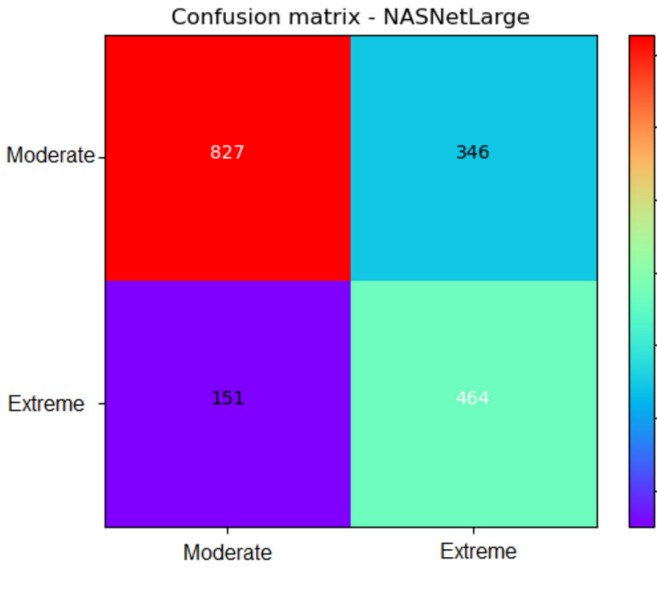

**Figure 1** Confusion matrix for the best NASNetLarge model. This NASNetLarge model corresponds to the one with data augmentation of horizontal flip (first column in the new model specification section of table 2). The figure shows the absolute number of images in each label: observed (true) on the y-axis and predicted on the x-axis.

Third, building on the last specification (ie, model with a decay of 1/10), we updated the monitoring factor which updated the learning rate when it did not improve through epochs. Originally, this factor was 0.1, and we updated it to 0.3. This model did not substantially improve the performance of the model.

### GradCam

In the GradCam (ie, class activation maps) analysis, we used the NASNetLarge model with one data augmentation technique (horizontal flip). Even though the performance of the NASNetLarge model with two data augmentation techniques (horizontal flip and zoom range) was better in more metrics, the model with horizontal flip only had better recall and F1 score for the extreme label. The main indications for a moderate risk classification were the presence of green areas and lack of close nearby buildings. That is, images with several open spaces such as parks, open streets or wide avenues would most likely be classified as moderate risk. Conversely, areas close to buildings, with a considerable presence of people, and with meeting points (eg, street vendors) were often classified as extreme COVID-19 risk. In other words, bus stops with one or multiple street vendors, newspapers stand or any other point for people to gather around would most likely be classified as extreme risk. The presence of cars did not seem to impact the classification process.

## DISCUSSION
### Main findings

With almost all research on computer vision and COVID-19 focusing on diagnostic models based on X-rays and other clinical images, our work is novel because it borrows techniques from computer vision into epidemiology and population health leveraging on available data (street images). In this study, we showed that deep CNNs can classify street images according to their COVID-19 risk with acceptable accuracy. Future work should strengthen available CNNs or develop a new architecture which could maximise the accuracy classification, not only for a binary outcome but also covering multiple outcomes. This work could spark interest to use CNNs—and other artificial intelligence tools—to advance population health and the epidemiological knowledge of COVID-19 (and other diseases), above and beyond the applications of CNNs for diagnosis and prognosis of individual patients (eg, classification of X-rays and compute tomography images of the chest[1–3]).

### Results in context

This work signalled that a deep neural network is moderately accurate to classify street images according to COVID-19 risk levels. These results are encouraging because the task we pursued was difficult: to classify street images into levels for which there is no unique intrinsic information in the images. Classification of, for example, X-ray images of the chest into healthy or ill could be easier for a CNN because the X-ray of someone with a disease (eg, pneumonia) would have unique features (eg, infiltrate spots at the bottom of the lungs) that an X-ray of the chest of someone healthy would not have at all. Conversely, in our case, the street images did not have a unique underlying pattern to guide the classification process. Our model had to work harder to find those unique characteristics to decide between moderate and extreme risk.

Further tuning of the selected model (NASNetLarge) suggested that data augmentation methods improved the performance of the model. When we updated the learning rate optimiser (decay and factor parameters), the model performance did not substantially improve. This could suggest that, for this particular task, we may need a large number of images. Alternatively, several combinations of data augmentation techniques would need to be tested. Data augmentations should be carefully considered to select those most suitable for these images; for example, vertical flip may not be a reasonable choice for street images.

Nguyen *et al* used Google Street View images to associate features of the built environment with COVID-19 cases in several states in the USA.[4] Although we could have followed the same approach, there would be some unique local features of the built environment that may not have been identified by available object detection tools (eg, street vendors and newspaper stands). We are not aware of other peer-reviewed papers in which street images have

been classified according to COVID-19 outcomes developing a new model or leveraging on transfer learning from an established neural network. Our work contributes to the available literature with a newly trained model benefiting from transfer learning from a large and well-known architecture (NASNetLarge), based on images from a city in an upper-middle-income country (Lima, Peru).

The activation maps (GradCam results) are not only useful to analyse the model's interpretation capability, but they also bolster the existing evidence of crowded places or indoor venues (such as nearby buildings) as COVID-19 high-risk areas. For example, areas with street vendors would activate more than open spaces for the extreme risk classification; on the other hand, open areas would play a major role in classifying moderate risk images. Overall, our findings agree with the evidence describing crowded areas, such as restaurants, gyms, hotels and cafes, as having high COVID-19 transmission risk.[14] Furthermore, our work advances the field by showing that street images with no other clinical or epidemiological data have moderate accuracy to predict COVID-19 risk.

### Public health implications

Our work could have pragmatic applications to better understand the epidemiology of COVID-19 and to inform public health interventions. For example, our model—and future work improving this analysis—could be used to characterise bus stops and other public places for which labelled data are not available. We worked with images from bus stops in Lima, and our model could be applied to bus stops in other cities to characterise their COVID-19 risk, particularly where observed data are not available. Furthermore, our work could spark interest to conduct more sophisticated analyses, like semantic segmentation whereby some unique elements of the local environment could be identified as potential high-risk places. For example, bus stops in Lima often host food street vendors and newspaper stands where people usually gather. Perhaps, the bus stops themselves are not high-risk places, but those surrounding shops. This could inform policies and interventions to reduce the COVID-19 risk in these places. Overall, deep learning techniques, including CNNs, could be adopted by epidemiological research to advance the evidence about risk factors as well as disease outcomes and distribution, in addition to their current use in clinical medicine.[1–3]

Our work was designed to understand whether and how well street images, without complementary data, can predict COVID-19 risk. Our results support the idea that the built environment alone is a health determinant because the street images were not complemented with other epidemiological data such as number of cases or COVID-19 transmission. Measuring COVID-19 throughout a country can be challenging and barriers include lack of access to tests as well as laboratory facilities to process the samples, and limited health or trained personnel to take the samples. Our work suggests

that street images could serve as proxy to estimate the COVID-19 risk in places where this information does not exist based on observed data. Therefore, we provide preliminary evidence suggesting that street images can be instrumental in COVID-19 surveillance.

Finally, as argued before, this is a pilot feasibility proof-of-concept study to study whether CNNs could classify street images according to COVID-19 indicators. This work complements the current use of CNNs for COVID-19 classification of clinical images (eg, X-rays). This work should be regarded as the first step in the use of CNNs in epidemiology and population health relevant to COVID-19; this work is not the ultimate work on this subject and future research should improve our approach and results.

### Ongoing and future work

Ongoing and future work includes the development of a classification model for the four outcome labels (ie, moderate, high, very high and extreme COVID-19 risk). We will implement techniques that can potentiate the classification capacity of the neural networks, including ensemble models,[15] novel loss functions not currently implemented in the Keras environment (eg, squared earth mover's distance-based loss function),[16] and we may try other architectures (eg, SqueezeNet[17]) with similar precision yet less computationally expensive. Because most of our bus stop images also depicted buildings, we may try to use a network already trained on images of buildings and other city landscapes (eg, Places-365).

### Strengths and limitations

We followed a predefined protocol which included transfer learning leveraging on large and deep neural networks trained with millions of images (ImageNet). We still had to train the parameters of the output layer, for which we did not have a massive number of images. Future work could expand our analysis with information and images from more bus stops or other public spaces to train a more robust model. Ideally, these images should come from different cities. This information may be available in other countries. There are further limitations we must acknowledge. First, the images and labels were not synchronic; that is, the figures and the labels were not collected on the same date. This is a shared limitation with other studies working with street images from open sources (eg, Google Street View), because these images are not taken continuously or in real time. This should not be a major limitation because the analysis mostly focused on the built environment, which has not changed substantially in recent years. Because this feasibility study showed that the classification model performed moderately well, researchers could collect new images in a prospective work to verify our findings with synchronic data. In this line, satellite images collected daily could be useful. Second, we did not have exact details on how the bus stops were classified by the local authorities. Nevertheless, we used official information which is provided to the public for their safety and

to inform them about the progression of the COVID-19 pandemic. Because it is an official source of public information, we trust their method for classification is sound and based on the best available evidence. This limitation should not substantially bias our model or results because the labels were clearly available from the data provider (transport authority), and we did not have to make any assumptions nor manual labelling. However, this may limit the external reproducibility of our work because other researchers may not label their images following the same criteria by our data source. We argue that this should not rest importance to our work because which could serve as basis for future research in the area in which the underlying labelling criteria are clearer. Third, we had five images per bus stop: the fifth image did not look at a specific angle, unlike the other four images that looked at 0°, 90°, 180° and 270° around the bus stop. Therefore, the fifth image had some overlap with the other images. We took this decision to maximise the available data. Researchers with access to more labelled information, perhaps from public places overseas, could use the four images without overlap and not significantly reducing the dataset size. In this line, the datasets (training, test and validation) were split randomly and, just by chance although improbably, all images of one particular bus stop could have fallen in a subset (eg, test dataset). If so, the model would have poor accuracy to predict this specific bus stop because the model did not have any information/images about that bus stop in the training dataset. However, because we trained the model to classify moderate and extreme risk of COVID-19, the model learnt patterns and profiles of the bus stops and their surrounding areas. This training could then be applied to other bus stops with similar characteristics. The GradCam analysis helped us to exemplify the patterns most influential in the selection process. Arguably, the influential patterns would be in all or most images. Fourth, our model cannot be independently reproduced because we could not make the underlying data available because these images do not belong to us. Google Street View images are available through the API, though they need personal login credentials. Although this would not replace the raw underlying data, to increase the transparency of our work we made available the Jupyter notebooks used in the analysis (online supplemental materials). These notebooks show the codes and results. Fifth, we did not report or discuss the algorithms or computations behind the CNNs we used for transfer learning. As per our protocol, we chose and applied a set of established CNNs to solve a classification problem. Disentangling the underlying mechanisms underneath each CNN was beyond the scope of this work. Nevertheless, it is relevant to understand the areas of the images most influential in the classification process. This way, we can verify if the classification process followed a logical path. We therefore reported the GradCam analysis.

## Conclusions

This study showed that a CNN has moderate accuracy to classify street images into moderate and extreme risk of COVID-19. In addition to applications in clinical medicine, deep CNNs have the potential to also advance the epidemiology of COVID-19 at the population level exploding unstructured and non-conventional data sources.

**Contributors** RMC-L and JFHSC conceived the idea. RMC-L conducted the analysis with support from JFHSC and MC-C. MC-C supported the revised analysis. All authors approved the submitted version. RMC-L is the guarantor for this study.

**Funding** RMC-L is supported by a Wellcome Trust International Training Fellowship (Wellcome Trust 214185/Z/18/Z).

**Competing interests** None declared.

**Patient and public involvement** Patients and/or the public were not involved in the design, or conduct, or reporting, or dissemination plans of this research.

**Patient consent for publication** Not applicable.

**Ethics approval** Not applicable.

**Provenance and peer review** Not commissioned; externally peer reviewed.

**Data availability statement** Data may be obtained from a third party and are not publicly available. Outcome (ie, labels: moderate and extreme COVID-19 risk) data are available online: https://sistemas.atu.gob.pe/paraderosCOVID; this information was systematised at https://github.com/jmcastagnetto/lima-atu-covid19-paraderos. The images were downloaded from Google Street View through the API with a personal account; images cannot be shared with third parties. All analysis codes are available as Python Jupyter Notebooks in the online supplemental materials. JupyterLab Notebooks and the final model (weights) are available at: https://figshare.com/articles/online_resource/Street_images_classification_according_to_COVID-19_risk_in_Lima_Peru_A_convolutional_neural_networks_feasibility_analysis/17321021.

**ORCID iDs**
Rodrigo M Carrillo-Larco http://orcid.org/0000-0002-2090-1856
Manuel Castillo-Cara http://orcid.org/0000-0002-2990-7090

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
