## [Reviewer comments · BMJ Open]

ARTICLE DETAILS

TITLE (PROVISIONAL)	Street images classification according to COVID-19 risk in Lima, Peru: A convolutional neural networks feasibility analysis
AUTHORS	Carrillo-Larco, Rodrigo; Castillo-Cara, Manuel; Hernández Santa Cruz, Jose Francisco

VERSION 1 – REVIEW

REVIEWER	Jayakrishnan Ajayakumar Case Western Reserve University, School of Medicine
REVIEW RETURNED	20-May-2022

GENERAL COMMENTS	1) This paper should focus more on the public health impact rather than the machine learning aspect. Currently the paper seems to be just focused on CNN rather than Covid-19 risk. 2) Introduction is rather very small and the authors have left out many works that have utilized Google Street Imagery along with ML to classify health risks. (For eg. Nguyen QC, Huang Y, Kumar A, Duan H, Keralis JM, Dwivedi P, Meng HW, Brunisholz KD, Jay J, Javanmardi M, Tasdizen T. Using 164 Million Google Street View Images to Derive Built Environment Predictors of COVID-19 Cases.) 3) "However, there is no evidence on convolutional neural networks being used for population health" (this is a very strong statement to make and there are many a papers that uses ML approaches (CNN being just one of them)) to solve population health challenges. 4) The selection of just binary label seems to be based on convenience. There is no justification for such a decision. 5) There should be a brief description about how the transport department has classified the images. Then it would be interesting to compare the key features that were used by the department for the classification and the features identified by the Neural Network. 6) Ours is a proof-of-concept work (This is mentioned in multiple places which is unnecessary) 7) "The main indications for a moderate risk classification were the presence of green areas and lack of close nearby buildings (Figure 2). Areas close to buildings or with a considerable presence of people were often classified as extreme COVID-19 risk. The presence of cars did not seem to impact the classification" This is the main section of the paper and needs to be elaborated. The presence of people could be a key indicator, but this is currently poorly explained and even the Gradcam diagrams doesn't clarify
--

	anything concrete. 8) As previously mentioned, as this is a supervised algorithm the details about labelling step is key. If a bustop is classified as extreme are the 5 images related to it classified as extreme. Such details should be provided and its imperative that the labelling strategy should be clearly explained (eventhough it is done by the transport department).
REVIEWER	Taha E. Taha Menoufia University
REVIEW RETURNED	26-Jun-2022
GENERAL COMMENTS	- Implementation procedures and results have been well-reported - Both the methodology used and flow of ideas are ok - Contribution of this work is good;

VERSION 1 – AUTHOR RESPONSE

Reviewer #1

C1. This paper should focus more on the public health impact rather than the machine learning aspect. Currently the paper seems to be just focused on CNN rather than Covid-19 risk.

R1. We have further elaborated on the public health relevance of our work. Please, refer to:

- Introduction section (p. 04 of tracked version) which was edited to elaborate on the public health implications of our work in a clearer fashion.
- Methods section, “Rationale” sub-heading (pp. 05-06 of tracked version), in which we included a new paragraph: *“Public health and epidemiological research usually rely on structured data sources such as health surveys and measurements from patients including samples such as blood. Unstructured data sources, such as images, are gaining attention in clinical medicine and have been used to develop diagnosis and prognostic models; however, the use of images, including street images, in public health and epidemiological research is limited. This work elaborates on this premise and on the current burden by COVID-19 and was conceived to study whether street images can ascertain the COVID-19 risk in the community. If successful, a deep learning model to classify street images according to COVID-19 risk could be used for disease surveillance, and to estimate the risk in places where observed data lack.”*
- Discussion section, “Public health implications” sub-heading (p. 12 of tracked version), in which we included a new paragraph: *“Our work was designed to understand whether and how well street images, without complementary data, can predict COVID-19 risk. Our results support the idea that the built environment alone is a health determinant because the street images were not complemented with other epidemiological data such as number of cases or COVID-19 transmission. Measuring COVID-19 throughout a country can be challenging and barriers include lack of access to tests as well as laboratory facilities to process the samples, and limited health or trained personnel to take the samples. Our work suggests that street images could serve as proxy to estimate the COVID-19 risk in places where this information does not exist based on observed data. Therefore, we provide preliminary evidence suggesting that street images can be instrumental in COVID-19 surveillance.”*

C2. Introduction is rather very small and the authors have left out many works that have utilized Google Street Imagery along with ML to classify health risks. (For eg. Nguyen QC, Huang Y, Kumar A, Duan H, Keralis JM, Dwivedi P, Meng HW, Brunisholz KD, Jay J, Javanmardi M, Tasdizen T. Using 164 Million Google Street View Images to Derive Built Environment Predictors of COVID-19 Cases.)

R2. We have further elaborated on the background literature in the Introduction section (p. 04 of tracked version).

“In a similar vein, though not exclusively addressing COVID-19, other researchers have leveraged on street images to study health-related social inequalities,¹ walkability,² air pollution,³ as well as the built environment and health outcomes.^{4,5} These examples show the potential of computer vision for population health research, above and beyond its multiple applications in clinical medicine with diagnostic and prognostic models.”

The suggested reference by Nguyen *et al.* had already been included, and we are including new citations where street images were used to study population health outcomes.

C3. "However, there is no evidence on convolutional neural networks being used for population health" (this is a very strong statement to make and there are many a papers that uses ML approaches (CNN being just one of them)) to solve population health challenges.

R3. This statement has been toned down, rewritten, and specified for COVID-19 alone. Please, refer to page 05 in the tracked version document.

“While there is evidence about convolutional neural networks being used for classification of X-rays and other clinical images for COVID-19 diagnosis,⁶⁻⁸ there is less evidence on convolutional neural networks being used for population health and COVID-19.”

C4. The selection of just binary label seems to be based on convenience. There is no justification for such a decision.

R4. This decision has been further justified (p. 06 of tracked version).

“We conducted a pilot feasibility study considering two outcome labels only. This, because we aimed to ascertain whether our hypothesis was possible and lead to relevant results while studying, from a public health perspective, the most important outcomes signalling the extremes of the risk distribution. Developing a model to identify areas at moderate risk could signal places where restrictions can be relaxed or suspended. Similarly, developing a model to identify areas at extreme risk would signal places where restrictions should be kept or strengthened. Therefore, a model focusing on two labels only, where these labels represent the extremes of the risk distribution, would be relevant and provide actionable evidence.”

C5. There should be a brief description about how the transport department has classified the images. Then it would be interesting to compare the key features that were used by the department for the classification and the features identified by the Neural Network.

R5. Unfortunately, the exact methods or criteria followed by local authorities to classify bus stops are not available. In other words, how they defined a bus stop was at moderate, high, very high or extreme risk, was not provided. This has been acknowledged in the Methods section and discussed in the limitations.

[Methods - p. 06 of tracked manuscript] *“Although this is an official source of information from a government branch, details of how the bus stops were classified are not available; please, refer to the discussion section where we further elaborate on this caveat.”*

[Discussion - p. 19 of tracked manuscript] *“Nevertheless, we used official information which is provided to the public for their safety and to inform them about the progression of the COVID-19. Because it is an official source of public information, we trust their method for classification is sound and based on the best available evidence. This limitation should not substantially bias our model or results because the labels were clearly available from the data provider (transport authority), and we did not have to make any assumptions nor manual labelling. However, this may limit the external reproducibility of our work because other researchers may not label their images following the same criteria by our data source. We argue that this should not rest importance to our work because which*

could serve as basis for future research in the area in which the underlying labelling criteria are clearer.”

C6. Ours is a proof-of-concept work (This is mentioned in multiple places which is unnecessary)

R6. The manuscript has been edited throughout to remove this statement. The “proof-of-concept” term has been kept in two sentences only: on page 05 (Methods) and on page 12 (Discussion) of the tracked version.

C7. "The main indications for a moderate risk classification were the presence of green areas and lack of close nearby buildings (Figure 2). Areas close to buildings or with a considerable presence of people were often classified as extreme COVID-19 risk. The presence of cars did not seem to impact the classification" This is the main section of the paper and needs to be elaborated. The presence of people could be a key indicator, but this is currently poorly explained and even the Gradcam diagrams doesn't clarify anything concrete.

R7. We further elaborated on the Gradcam analysis in different sections of the revised manuscript.

[Methods - p. 08 of tracked version] *“Areas most activated as shown by brighter colours, would be decisively in the classification process.”*

[Results - p. 10 of tracked version] *“The main indications for a moderate risk classification were the presence of green areas and lack of close nearby buildings. That is, images with several open spaces like parks, open streets, or wide avenues, would most likely be classified as moderate risk. Conversely, areas close to buildings, with a considerable presence of people, and with meeting points (e.g., street vendors), were often classified as extreme COVID-19 risk. In other words, bus stops with one or multiple street vendors, newspapers stand, or any other point for people to gather around, would most likely be classified as extreme risk. The presence of cars did not seem to impact the classification process.”*

[Discussion - p. 12 of tracked version] *“The activation maps (GradCam results) are not only useful to analyse the model's interpretation capability, but they bolster the existing evidence of crowded places or indoor venues (such as nearby buildings) as COVID-19 high-risk areas. For example, areas with street vendors would activate more than open spaces for the extreme risk classification; on the other hand, open areas would play a major role in classifying moderate risk images. Overall, our findings agree with the evidence describing crowded areas, such as restaurants, gyms, hotels, and cafes, as having high COVID-19 transmission risk.¹⁵ Furthermore, our work advances the field by showing that street images with no other clinical or epidemiological data, have moderate accuracy to predict COVID-19 risk.”*

C8. As previously mentioned, as this is a supervised algorithm the details about labelling step is key. If a bustop is classified as extreme are the 5 images related to it classified as extreme. Such details should be provided and its imperative that the labelling strategy should be clearly explained (even though it is done by the transport department).

R8. The fact that it is unknown how exactly each bus stop was labelled by the local transport authority is acknowledged in the Methods and Discussion sections. Please, refer to our fifth answer above for further details on how we have addressed this comment raised by the reviewer.

Whether the surrounding area to the bus stop received the same label as the index bus stop has been detailed in the Methods section (p. 06 tracked version).

“Consequently, if the bus stop was labelled as moderate or extreme risk, the same label applied to the images of the surrounding area. For example, if bust stop X was labelled as moderate risk, all five images for such bus stop were labelled as moderate risk (i.e., image of the bus top itself plus the four images of the surrounding area).”

Reviewer #2

C1. Comments to the Author:

- Implementation procedures and results have been well-reported**
- Both the methodology used and flow of ideas are ok**
- Contribution of this work is good**

R1. We appreciate the positive feedback by this reviewer. We are glad the reviewer found merit in our work.